

# Radiative absorption enhancement of dust mixed with anthropogenic pollution over East Asia

Pengfei Tian[1,2], Lei Zhang[1], Jianmin Ma[2,3], Kai Tang[1], Lili Xu[1], Yuan Wang[4], Xianjie Cao[1], Jiening Liang[1], Yuemeng Ji[5], Jonathan H. Jiang[6], Yuk L. Yung[4], Renyi Zhang[5]

[1]Key Laboratory for Semi-Arid Climate Change of the Ministry of Education, College of Atmospheric Sciences, Lanzhou University, Lanzhou, China

[2]Key Laboratory for Environmental Pollution Prediction and Control, Gansu Province, College of Earth and Environmental Sciences, Lanzhou University, Lanzhou, China

[3]Laboratory for Earth Surface Processes, College of Urban and Environmental Sciences, Peking University, Beijing, China

[4]Division of Geological and Planetary Sciences, California Institute of Technology, Pasadena, CA 91125, USA

[5]Department of Atmospheric Sciences, Texas A&M University, College Station, Texas, 77843, USA

[6]Jet Propulsion Laboratory, California Institute of Technology, Pasadena, California 91125, USA

Correspondence to: Lei Zhang (zhanglei@lzu.edu.cn) and Yuan Wang (Yuan.Wang@caltech.edu)



**ABSTRACT:** The particle mixing state plays a significant yet poorly quantified role
in aerosol radiative forcing, especially for the mixing of dust (mineral absorbing) and
anthropogenic pollution (black carbon absorbing) over East Asia. We have
investigated the absorption enhancement of mixed-type aerosols over East Asia by
using the Aerosol Robotic Network observations and radiative transfer model
calculations. The mixed-type aerosols exhibit significantly enhanced absorbing ability
than the corresponding unmixed dust and anthropogenic aerosols, as revealed in the
spectral behavior of absorbing aerosol optical depth, single scattering albedo, and
imaginary refractive index. The aerosol radiative efficiencies for the dust, mixed-type,
and anthropogenic aerosols are -101.0, -112.9 and -98.3 $Wm^{-2}\tau^{-1}$ at the bottom of the
atmosphere (BOA), -42.3, -22.5 and -39.8 $Wm^{-2}\tau^{-1}$ at the top of the atmosphere
(TOA), and 58.7, 90.3 and 58.5 $Wm^{-2}\tau^{-1}$ in the atmosphere (ATM), respectively. The
BOA cooling and ATM heating efficiencies of the mixed-type aerosols are
significantly higher than those of the unmixed aerosol types over the East Asia region,
resulting in atmospheric stabilization. In addition, the mixed-type aerosols correspond
to a lower TOA cooling efficiency, indicating that the cooling effect by the
corresponding individual aerosol components is partially counteracted. We conclude
that the interaction between dust and anthropogenic pollution not only represents a
viable aerosol formation pathway but also results in unfavorable dispersion conditions,
both exacerbating the regional air pollution in East Asia. Our results highlight the
necessity to accurately account for the mixing state of aerosols in atmospheric models
over East Asia, in order to better understand the formation mechanism for regional air
pollution and to assess its impacts on human health, weather, and climate.





# 1 Introduction

Atmospheric aerosols or particulate matter (PM) profoundly affect the energy budget of the earth-atmosphere system directly by interfering with the radiative transfer and indirectly by modifying cloud formation (Twomey, 1977; Charlson and Schwartz, 1992; Fan et al., 2007; Wang et al., 2011). However, the assessment of the aerosol radiative effects is limited because of the inherent difficulties associated with observations and model simulations (Stevens and Bony, 2013; Kok et al., 2017). In particular, an accurate quantification of the mixing state of absorbing aerosols poses a great challenge in the estimation of the aerosol direct radiative forcing (Haywood and Boucher, 2000; He et al., 2015). Currently, the assessment of the aerosol direct and indirect radiative forcing represents large uncertainty in the prediction of future climate by anthropogenic activities (IPCC, 2007; IPCC, 2013). Moreover, aerosol mixing state significantly affects atmospheric dynamics (Ramanathan and Carmichael, 2008).

Atmospheric aerosols are typically internally and/or externally mixed during their lifetimes (Jacobson, 2001; Zhang and Zhang, 2005; Zhang et al., 2008; Khalizov et al., 2009a; Pagels et al., 2009; Taylor et al., 2015). The East Asian region is experiencing persistent heavy air pollution conditions in the present day (Guo et al., 2014; Zhang et al., 2015; Wang et al., 2016). Black carbon (BC) is one of the major anthropogenic pollutants in this region that exerts significant environmental and climatic effects because of the strong absorption of solar radiation (Wang et al., 2012; Peng et al., 2016). East Asia is also the largest dust source region second to the Saharan Desert (Huang et al., 2014; Huang et al., 2015; Tian et al., 2015). As a result, coarse mode dust particles are frequently mixed with anthropogenic pollution along their transport pathway in East Asia (Noh et al., 2012; Logan et al., 2013; Guo et al.,



2017; Hara et al., 2017). The potential anthropogenic influence on dust has been
investigated close to the dust source regions (Huang et al., 2010; Bi et al., 2017).
Lower single scattering albedo (SSA) of mixed dust plumes has been assessed in
previous studies (Kim et al., 2005; Khatri et al., 2014). Li et al. (2015) have studied
the SSA spectral curvature of the East Asian aerosol mixtures using Aerosol Robotic
Network (AERONET) products and model simulations. The observations from a
sun–sky radiometer and a lidar have been applied to identify the presence of Asian
dust in mixed aerosol plumes at several East Asian monitoring sites (Noh et al., 2017).
Those earlier studies on optical properties and radiative effects of the East Asian dust
and anthropogenic aerosol mixtures have promised reduction of the uncertainties in
estimating the aerosol radiative effects.

Internal mixing of coarse mode dust with fine mode anthropogenic aerosols has

been suggested by observations in the Asian Aerosol Characterization Experiment
(ACE-Asia) (Seinfeld et al., 2004) and recent studies (Sugimoto et al., 2015; Wang et
al., 2017), although the mechanism leading to the mixing has yet to be elucidated.
Internal mixing of dust with anthropogenic pollution likely occurs via condensation of
low-volatility organic and inorganic compounds, particle-phase reactions, and
coagulation with other aerosol types (Zhang et al., 1996; Zhao et al., 2006; Qiu et al.,
2011). In addition, dust particles provide reactive surfaces for catalytic conversion of
sulfur dioxide to sulfate, which has been suggested as a key mechanism for severe
haze formation in China (Zhang et al., 2015; Li et al., 2017a). Atmospheric
measurements using electron microscopy have identified BC and certain soluble
aerosols on the surface of dust particles (Tobo et al., 2010; Ma et al., 2012; Li et al.,
2014). Pan et al. (2017) have studied the morphology change of East Asian dust
mixing with anthropogenic aerosols and showed the possibility evidence for the





occurrence of aqueous-phase reactions.

The aerosol mixing state significantly affects the radiative effects (Jacobson,

2001; Khalizov et al., 2009b; Xue et al., 2009). Several previous studies have shown
that the amount of solar radiation reaching the Earth surface through mixtures of
mineral dust and other absorbing aerosols is considerably reduced compared to that
through dust-only aerosols (Derimian et al. 2008; Obregón et al. 2015). Researchers
have reported that the radiative efficiency of non-dust aerosols is higher than that of
dust aerosols at two urban Asian cities of Gwangju and Beijing (Noh et al., 2012; Yu
et al., 2016). Maximum radiative efficiency under unpolluted conditions has been
found by comparing aerosol radiative effects under different air quality conditions
(Chen et al., 2016). Aerosol radiative efficiency has been found to be strongly
influenced by aerosol absorbing ability (e.g., SSA) and size of fine mode particles in
Central China (Zhang et al., 2017).

Although the mixing state of dust and anthropogenic aerosols considerably

affects aerosol radiative effects, the radiative absorption enhancement by the aerosol
mixtures in East Asia has not been assessed. In this present work, we have extensively
investigated the radiative absorption enhancement by the East Asian aerosol mixtures
on the basis of long-term AERONET observations and SBDART model simulations.
We classified the dust, mixed-type, and anthropogenic aerosols according to the
aerosol SSA spectral behavior and Ångström exponent parameter. The optical and
microphysical properties of the various aerosol types were analyzed, with emphasis
on the absorption enhancement by mixed-type aerosols. The mechanism leading to the
radiative absorption enhancement by the mixed-type aerosols and their impacts on
regional air pollution and climate have been discussed. Our results suggest that the
East Asian aerosol mixtures result in a more stable atmosphere that is unfavorable for



diffusion and dispersion of the atmospheric pollutants.
**2   Data and methodology**
**2.1   AERONET data**
The aerosol optical and microphysical data used in this research were originally
from the Aerosol Robotic Network (AERONET) (Holben et al., 1998). To ensure the
data quality, we only analyzed the cloud-screened, quality-assured Level 2.0 inversion
data. We further constrained the data with solar zenith angle between $50\,^\circ$ and $80\,^\circ$ to
avoid possible inversion errors and surface albedo smaller than 0.5 to exclude
seasonal snow-covered surfaces. The AERONET spectral products are available at the
wavelengths of 440, 675, 870, and 1020 nm, respectively. Previously, Dubovik et al.
(2000) have evaluated the uncertainty of the AERONET products.
All available worldwide measurement sites from the Aerosol Robotic Network
(AERONET) program with a sample number of greater than 100 were included in the
present work. Considering the regional representativeness and data availability, 11
sites (Table S1 and Fig. S1) were selected to represent the various types of the East
Asian aerosol mixtures. For example, the sample numbers of Beijing and Xianghe
were randomly reduced to a one-fourth of the total numbers because these two sites
are close to each other and the total sample numbers of the two sites are significantly
larger than the others.
**2.2   Radiative forcing and efficiency calculations**
The aerosol direct radiative forcing ($\Delta F$) is defined as follows:
$$\Delta F = (F^{\downarrow} - F^{\uparrow}) - (F_0^{\downarrow} - F_0^{\uparrow}) \tag{1}$$

where $F$ and $F_0$ are radiative fluxes under the aerosol-free and aerosol-laden
conditions respectively, the upward and downward arrows denote the directions of the
radiative fluxes. The AERONET products include the aerosol radiative forcing and

efficiency, in addition to the optical and microphysical parameters. However, the radiative forcing in the AERONET products was not exactly calculated using Equation (1). Instead, we calculated the aerosol direct radiative forcing and efficiency by using the widely adopted Santa Barbara DISORT Atmospheric Radiative Transfer (SBDART) model (Ricchiazzi et al., 1998). The aerosol radiative efficiency ($\Delta F^{eff}$) is defined as the aerosol direct radiative forcing per unit aerosol optical depth (AOD):

$$\Delta F^{eff} = \Delta F / AOD_{0.55} \tag{2}$$

where $AOD_{0.55}$ is the 550 nm AOD. However, the aerosol direct radiative forcing is not linearly dependent on AOD (e.g., Wu et al., 2015). To exclude possible errors from aerosol loading in calculating the aerosol radiative efficiency in Equation (2), the 550 nm AOD was set to unity ($AOD_{0.55}=1.0$) in SBDART model calculations. The data processing procedure is presented in Fig. 1 and a detailed discussion of aerosol radiative forcing and efficiency calculation has been provided elsewhere (Tian et al., 2018).

**2.3 Aerosol classification**

The SSA spectral behavior and the Ångström exponent parameter were applied to identify aerosol mixtures (Fig. 1). The main absorbing component of the anthropogenic pollutants is BC, which exhibits strong absorption throughout a broad wavelength range (Bond and Bergstrom, 2006). Dust aerosols are also an absorbing medium because of the iron-bearing minerals such as hematite, goethite and magnetite, which enable the dust aerosols to absorb strongly in the UV and short visible wavelengths and weakly in the near infrared (Schuster et al., 2016). Thus, the dust aerosols exhibit a monotonically increasing SSA trend with increasing wavelength, while the anthropogenic pollutants (the absorption of which is mainly contributed by BC) show a monotonically decreasing SSA spectra (Bergstrom et al., 2007; Giles et



169 al., 2012; Tian et al., 2017). The absorption of aerosol mixtures is contributed by both

170 dust and BC, leading to a non-monotonic SSA spectra (Khatri et al., 2014). The

171 characteristic non-monotonic SSA spectral behavior of mixed-type aerosols provides

172 a useful approach to identify the mixed-type aerosols. SSA curvature of greater than

173 0.1, which has been suggested for "moderate mixing" (Li et al., 2015), was applied

174 for the mixed-type aerosol. We employed an additional criterion, Ångström exponent,

175 which is related to the aerosol size and the widely used in aerosol classification (Eck

176 et al., 2010; Derimian et al., 2016), to further constrain the aerosol classification.

177 Results show that Ångström exponent values are smaller for coarse mode dust

178 aerosols (lower than 0.7 in the present study), larger for fine mode anthropogenic

179 aerosols (greater than 1.4 in this research), and range from 0.8 and 1.3 for the

180 mixed-type aerosols, respectively.

181  The imaginary refractive index at 440 nm ($k_{440}$, in the visible bands) versus the

182 average of the imaginary refractive indices at 675, 870, and 1020 nm ($k_{rnir}$, in the red

183 and near-infrared bands) is shown in Fig. 2. Anthropogenic aerosols with the

184 absorption mainly contributed by BC particles exhibit a flat imaginary refractive

185 index (Bond and Bergstrom, 2006), while dust with the components such as hematite

186 absorbs strongly in UV but weakly in longer wavelengths (Hsu and Matijević, 1985;

187 Schuster et al., 2016). Hence, the data of anthropogenic aerosols are scattered along

188 the 1:1 line (Fig. 2). Dust aerosols show stronger absorption at the visible wavelength

189 than at the red and near-infrared wavelengths, so the data are scattered above the 1:1

190 line. The data for most of the mixed-type aerosols lie above the 1:1 line and on the

191 right side of the $k_{rnir} = 0.0042$ threshold, which is suggested by Schuster et al. (2016)

192 to separate dust ($k_{rnir} < 0.0042$) and biomass burning ($k_{rnir} > 0.0042$) aerosols. The



mixed-type aerosols show the strongest absorption and most of the sites with a
mixed-type aerosol sample number of 50 and higher are located in East Asia. Hence,
our method quantitatively separated various aerosol types.

**3  Spectral behavior of the East Asian aerosol mixtures**

Extensive investigations of the spectral optical properties were carried out to
discuss the enhanced absorbing ability of the East Asian aerosol mixtures. Dust
aerosols show the highest spectral AOD and the anthropogenic aerosols exhibit the
lowest spectral AOD (Fig. 3a). The spectral dependence of dust aerosols is nearly
invariant across the wavelength spectrum while the dependence of anthropogenic
aerosols is relatively high, which is relevant to aerosol size and the Ångström
parameter (Ångström, 1929). The mixed-type aerosols have smaller AOD than dust
aerosols, but higher absorption aerosol optical depth (AAOD) throughout the
wavelength band of 440 to 1020 nm (Fig. 3b). Dust aerosols exhibit higher AAOD at
the 440 nm wavelength than that of anthropogenic aerosols.
As expected by using our SSA spectral classification method, dust aerosols show
a monotonic increasing SSA trend with increasing wavelength, while anthropogenic
aerosols exhibit an opposite trend. In contrast, the SSA of the mixed-type aerosols
peaks at the wavelength of 675 nm (Fig. 3c). Interestingly, the mixed-type aerosols
exhibit the lowest SSA value that cannot be predicted by our classification method
because the classification method considers the spectral trend rather than the value of
SSA. This indicates enhanced absorption for the mixed-type aerosols. The spectral
average SSA of the dust, mixed-type, and anthropogenic aerosols are 0.94, 0.89, and
0.93, respectively. Hence, internal mixing of the East Asian aerosol mixtures yields
the lowest SSA that is distinct from the corresponding individual aerosol types.
The dust aerosols exhibit the highest value for the real part of the complex



refractive index, while the anthropogenic aerosols show the lowest value (Fig. 3e).
Interestingly, the real refractive index of the mixed-type aerosols is close to that of
dust aerosols, indicating high scattering of the mixed-type aerosols. The spectral
imaginary refractive index for anthropogenic aerosols is nearly constant (Fig. 3f),
which is characteristic of this aerosol type. However, the imaginary refractive index
of anthropogenic aerosols is much lower than that of BC aerosols (approximately 0.6
in Bond and Bergstrom (2006)), because the majority of anthropogenic aerosol
components are non-absorbing aerosols such as sulfate and nitrate salts. The spectral
imaginary refractive index of dust aerosols is similar to the AERONET dust
climatology over Africa and the Middle East (Schuster et al., 2016), with stronger
absorption at the visible wavelength (440 nm). The mixed-type aerosols exhibit the
highest imaginary refractive index, especially at the visible wavelength, where the
imaginary refractive index of the mixed-type aerosols (0.0159) is more than twice that
of anthropogenic aerosols (0.0078).
Hence, the absorbing ability of the East Asian aerosols is significantly enhanced
due to the mixing process, in light of the lowest SSA and the highest AAOD and
imaginary refractive index for the mixed-type aerosols.
**4    Enhanced radiative absorption by East Asian aerosol mixtures**
To investigate the radiative effects caused by the enhanced absorbing ability of
the East Asian aerosol mixtures, we calculated the average aerosol direct radiative
efficiency at the bottom of the atmosphere (BOA), at the top of the atmosphere (TOA),
and in the atmosphere (ATM) (Fig. 4), respectively. The mixed-type aerosols exhibit
lower spectral average AOD (0.48) than dust aerosols (0.64) (Fig. 4a), but show
comparable radiative forcing relative to dust aerosols at BOA (-72.7 and -73.6 Wm$^{-2}$
for the mixed-type and dust aerosols, respectively) (Fig. 4a). This feature is explained





by higher BOA cooling efficiency of the mixed-type aerosols (-112.9 $Wm^{-2}\tau^{-1}$) than
dust (-101.0 $Wm^{-2}\tau^{-1}$) (Fig. 4b). The radiative absorption enhancement is evident for
the TOA and ATM forcing: the mixed-type aerosols exhibit the highest ATM radiative
forcing (55.0 $Wm^{-2}$) and the lowest absolute TOA forcing (-17.8 $Wm^{-2}$). For
comparison, we calculated the aerosol radiative efficiency of various aerosol types to
rule out the effect of aerosol loading. The mixed-type aerosols exhibit the highest
BOA cooling efficiency, the highest ATM heating efficiency (90.3 $Wm^{-2}\tau^{-1}$), and the
lowest TOA cooling efficiency (-22.5 $Wm^{-2}\tau^{-1}$).
The average BOA radiative efficiency of dust aerosols (-101.0 $Wm^{-2}\tau^{-1}$) in the
present study (Fig. 4b) falls in the range of -96.1 to -127.0 $Wm^{-2}\tau^{-1}$ by Yu et al. (2016),
but lower than the result of -124.6 $\pm$ 12.2 $Wm^{-2}\tau^{-1}$ by Noh et al. (2012). Note that
results by Noh et al. (2012) and Yu et al. (2016) are likely biased due to a non-linear
dependence between aerosol direct radiative forcing and AOD, which is avoided in
the present study. Our previous work (Tian et al., 2018), which used the same
radiative efficiency calculation method but a different aerosol classification approach
from the present study, obtained a similar BOA radiative efficiency of -102.3 $Wm^{-2}\tau^{-1}$
at the Semi-Arid Climate and Environment Observatory of Lanzhou University
(SACOL), Northwest China.
The spatial distributions of the aerosol radiative efficiency for BOA, ATM, and
TOA are presented in Figs. 5-7, respectively. Only those sites with a sample number
of greater than 50 were averaged and included in the figures. Despite the fact that
AERONET sites are unavailable in some remote areas, the worldwide aerosol
distributions are well captured by the AERONET observations. The mixed-type
aerosols are distributed in East Asia, India and around the Saharan Desert regions.
The mixed-type aerosols exhibit a higher BOA radiative cooling efficiency than



that of dust and anthropogenic aerosols over the East Asia region (Fig. 5). The BOA
radiative cooling efficiency over India is also high, but the difference of various
aerosol types is small. Biomass burning aerosols over Africa exhibit the highest BOA
cooling efficiency in the globe, which may be explained by the different combustion
compositions and processes as described in Eck et al. (2010) and Garc á et al. (2012).
Anthropogenic aerosols over North America and dust aerosols around the Sahara
Desert show a relatively lower BOA cooling efficiency.

The mixed-type aerosols also exhibit a higher ATM radiative heating efficiency

than dust and anthropogenic aerosols over East Asia (Fig. 6). The ATM radiative
heating efficiency over India is high for all aerosol types. The ATM radiative heating
efficiency is high over South Africa, where biomass burning aerosols dominate. On
the other hand, the ATM radiative heating efficiency over North America and around
the Sahara Desert regions is relatively smaller.

The enhanced BOA cooling and ATM heating efficiencies reveal that the

mixed-type aerosols exhibit higher BOA cooling and ATM heating effects than those
of the unmixed dust and anthropogenic aerosols with the same aerosol loading. The
enhanced BOA cooling and ATM heating effects lead to a cooler surface and warmer
atmosphere and restrain the development of the planetary boundary layer, resulting in
a more stable atmosphere that is unfavorable for dispersion of atmospheric gaseous
and PM pollutants. Noting that mixed-type aerosols occur frequently in East Asia and
their occurrence can reach as high as fifty percent over some locations (Li et al.,
2015). Hence, the mixed-type aerosols likely play a significant role in enhancing air
pollution over East Asia. In addition, the mixed-type aerosols show lower TOA
radiative cooling efficiency than dust and anthropogenic aerosols over the East Asia
region (Fig. S1). The reduced TOA cooling efficiency indicates that the East Asian



aerosol mixtures partially counteract the cooling effect of the Earth-atmosphere
system by the corresponding individual components.
**5 Discussions**
The aerosol direct radiative efficiency strongly depends on solar zenith angle
(e.g., Derimian et al., 2016). To investigate the influence of solar zenith angle on the
result of the present study, the aerosol radiative efficiency as a function of solar zenith
angle were calculated for various aerosol types (Fig. 7). Note that the AERONET data
used in the present study is only available between $50°$ to $80°$ solar zenith angles. The
BOA radiative cooling and ATM heating efficiencies (absolute value of radiative
efficiency) decrease with increasing solar zenith angle (decreasing cosine of solar
zenith angle), while the TOA cooling efficiency increase with increasing solar zenith
angle. The mixed-type aerosols exhibit higher BOA cooling efficiency, higher ATM
heating efficiency, and lower TOA cooing efficiency than those of unmixed dust and
anthropogenic aerosols. The dust aerosols exhibit both higher BOA and TOA cooling
efficiency than the anthropogenic aerosols, leading to small difference in the ATM
heating efficiency between the two aerosol types.
Aerosol absorption has been suggested as a key factor that determines the aerosol
radiative effects (Li et al., 2010). The aerosol radiative efficiencies as a function of
SSA and imaginary refractive index for various aerosol types were calculated to
investigate the effect of aerosol absorbing on the aerosols radiative efficiency (Figs. 8
and 9). The BOA cooling efficiency and ATM heating efficiency increase with
increasing absorption (i.e., decreasing SSA and increasing imaginary refractive index)
and the TOA cooling efficiency decrease with increasing absorption. However, the
dependences between radiative efficiency and SSA are stronger than those between
radiative efficiency and the imaginary refractive index for BOA, TOA, and ATM. The





dependences between radiative efficiency and SSA are approximately linear, but the

dependences between radiative efficiency and imaginary refractive index become less

apparent with increasing imaginary refractive index. The strong dependence between

aerosol radiative efficiency and SSA has also been shown over Central China and

desert and semi-desert regions of northwestern China (Xin et al., 2016; Zhang et al.,

2017).

We also examined the effects of the fraction of fine and coarse mode on aerosol

radiative efficiency (Fig. 10). The BOA cooling and ATM heating efficiencies (TOA

cooling efficiency) initially increase (decreases) with increasing fine mode fraction

(FMF), when FMF is lower than 0.3 and coarse mode dust aerosols dominate. Then

the BOA cooling and ATM heating efficiencies (TOA cooling efficiency) reach a peak

(bottom) in the FMF range of 0.3 to 0.5. Finally the BOA cooling and ATM heating

efficiency (TOA cooling efficiency) begin to decrease (increase) when FMF is greater

than 0.5, where fine mode anthropogenic aerosols become dominate. Overall, the

moderate mixing of dust with fine mode anthropogenic pollutants, which is classified

as mixed-type aerosols in the present study, is responsible for the enhanced radiative

absorption. A previous study (Li et al., 2015) has revealed that moderate mixing of

East Asian dust with fine mode pollutants is responsible for high SSA spectral

curvature, which suggests well-mixed aerosol mixtures.

**6 Conclusions**

The mixing state of atmospheric aerosols plays a significant yet poorly

quantified role in determining the aerosol optical properties and radiative effects. In

the East Asia region, coarse mode dust and fine mode anthropogenic pollution are

typically mixed externally and/or internally in the atmosphere. The mixing of dust

with anthropogenic aerosols exerts a significant influence on aerosol absorption and



radiative efficiency. We present an extensive investigation of the radiative effects of
the East Asian aerosol mixtures.

The mixed-type aerosols exhibit significantly higher AAOD, lower SSA, and

higher imaginary refractive index than those of unmixed dust and anthropogenic
aerosols, showing significantly enhanced absorption. The absorption enhancement is
most evident at wavelength 440 nm, where the imaginary refractive index of the
mixed-type aerosols (0.0159) is more than twice that of anthropogenic aerosols
(0.0078). The mixed-type aerosols also exhibit a unique non-monotonic SSA trend,
which provides a characteristic signature for identifying these aerosols.

The values of the aerosol radiative efficiencies for dust, mixed-type and

anthropogenic aerosols are -101.0, -112.9 and -98.3 $Wm^{-2}\tau^{-1}$ for BOA, -42.3, -22.5
and -39.8 $Wm^{-2}\tau^{-1}$ for TOA, and 58.7, 90.3 and 58.5 $Wm^{-2}\tau^{-1}$ for ATM, respectively.
The mixed-type aerosols exhibit significantly higher BOA radiative cooling efficiency
and ATM heating efficiency than those of dust and anthropogenic aerosols over East
Asia. These enhanced BOA cooling and ATM heating efficiencies reveal that the
mixed-type aerosols exhibit stronger BOA cooling and ATM heating effects than
those of unmixed dust and anthropogenic aerosols for a given aerosol loading,
resulting in a more stable atmosphere that is unfavorable for the diffusion and
dispersion of the gaseous and PM pollution. Hence, our results suggest that the
mixed-type aerosols likely play a significant role in enhancing the air pollution in East
Asia, because the mixing of dust and anthropogenic aerosols occurs frequently in this
region. In addition, the mixed-type aerosols show lower TOA cooling efficiency,
indicating that the mixed-type aerosols partially counteract the cooling effect of the
earth-atmosphere system by the corresponding individual components. Since dust
particles play a catalytic role in the conversion of sulfur dioxide to sulfate (Zhang et



al., 2015; Li et al., 2017a), the interaction between dust and anthropogenic aerosols
also provides a possible mechanism for the observed efficient internal mixing and
enhanced radiative absorption over East Asia.
Multiple factors have been suggested to be responsible for severe air pollution in
East Asia, including the interaction between BC aerosols and the atmospheric
boundary layer (Ding et al., 2016; Peng et al., 2016; Li et al., 2017b), rapid secondary
aerosol formation during severe haze events (Wang et al., 2016), weakening of the
East Asian monsoon circulation (Wu et al., 2016), and climate change (Cai et al,
2017). Our results indicate that interaction between dust and anthropogenic pollution
does not only represent a plausible pathway for PM formation and internal mixing but
also results in unfavorable dispersion conditions (i.e., increased atmospheric stability),
both exacerbating regional air pollution in East Asia. Clearly, future studies are
necessary to more accurately assess the mixing state of aerosols in atmospheric
models, in order to better understand the formation mechanism for air pollution over
East Asian and to assess its impacts on human heath, weather, and climate (Zhang et
al., 2015).
**7 Data availability**
The original sun photometer data are available from the AERONET website
(http://aeronet.gsfc.nasa.gov/). The radiative flux data for the worldwide AERONET
sites calculated using the SBDART model and all data for the figures and table in the
present research are available from the authors upon request.
**Competing interests**. The authors declare that they have no conflict of interest.
**Acknowledgements**. This research was financially supported by National Natural
Science Foundation of China (41627807 and 41475008) and National Key R&D



Program of China (2016YFC0401003). Yuan Wang acknowledged the support from
NASA ROSES ACMAP. Yuemeng Ji was financially supported by National Natural
Science Foundation of China (41675122) and Science and Technology Program of
Guangzhou city (201707010188). The authors thanke the principal investigators and
staff for establishing and maintaining the AERONET sites used in this research. We
thanke Institute for Computational Earth System Science (ICESS), University of
California for providing the SADART model. Y. Wang, J.H. Jiang, and Y.L. Yung
acknowledge support by the Jet Propulsion Laboratory, California Institute of
Technology, under contract with NASA.

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






**Figure Captions**


**Figure 1.** Diagram of the radiative efficiency calculation and aerosol classification
**Figure 2.** The imaginary refractive index at 440 nm versus the average of the imaginary refractive
indices at 675, 870, and 1020 nm for dust, mixed-type, and anthropogenic aerosols over
worldwide AERONET sites. Only those sites with a sample number of 50 and higher were
averaged and shown in the figure.
**Figure 3.** Spectral behavior of (a) AOD, (b) AAOD, (c) SSA, (d) asymmetry factor, (e) real part of
the complex refractive index, and (f) imaginary part of the complex refractive index for dust,
mixed-type and anthropogenic aerosols averaged from East Asian sites.
**Figure 4.** (a) Radiative forcing and (b) radiative efficiency of the dust, mixed-type, and
anthropogenic aerosols averaged from East Asian sites.
**Figure 5.** Aerosol radiative efficiency at BOA: (a) dust aerosols, (b) anthropogenic aerosols, and
(c) mixed-type aerosols.
**Figure 6.** Aerosol radiative efficiency in ATM: (a) dust aerosols, (b) anthropogenic aerosols, and
(c) mixed-type aerosols.
**Figure 7.** Aerosol direct radiative efficiency as a function of the cosine of solar zenith angle for
the dust, mixed-type and anthropogenic aerosols: (a) BOA, (b) ATM, and (c) TOA.
**Figure 8.** Aerosol direct radiative efficiency as a function of SSA for the dust, mixed-type and
anthropogenic aerosols: (a) BOA, (b) ATM, and (c) TOA.
**Figure 9.** Aerosol direct radiative efficiency as a function of imaginary refractive index for the
dust, mixed-type and anthropogenic aerosols: (a) BOA, (b) ATM, and (c) TOA.
**Figure 10.** Aerosol direct radiative efficiency as a function of FMF in the East Asian region: (a)
BOA, (b) ATM, and (c) TOA. The average radiative efficiencies of the dust, mixed-type and
anthropogenic aerosols are also plotted.







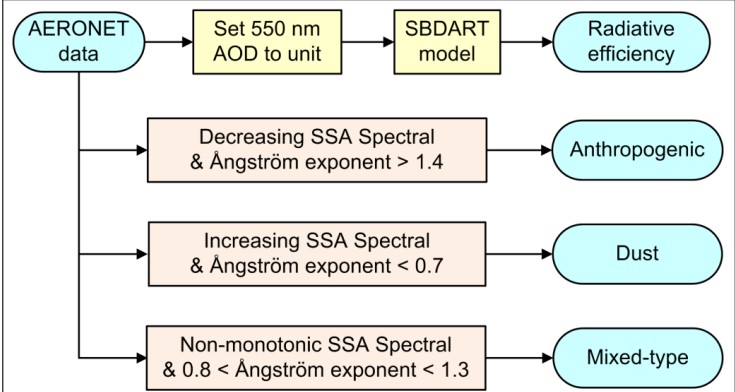


**Figure 1.** Diagram of the radiative efficiency calculation and aerosol classification




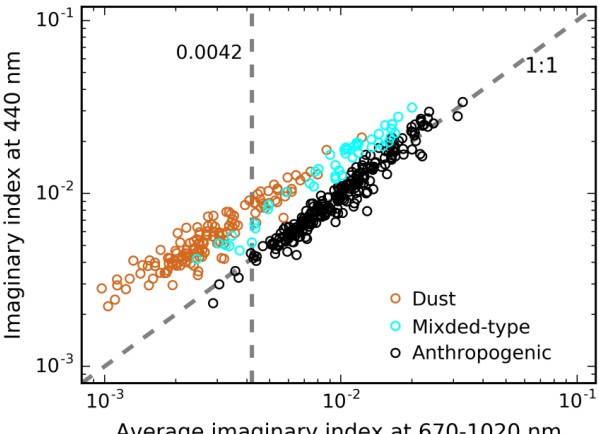


**Figure 2.** The imaginary refractive index at 440 nm versus the average of the imaginary refractive

indices at 675, 870, and 1020 nm for dust, mixed-type, and anthropogenic aerosols over

worldwide AERONET sites. Only those sites with a sample number of 50 and higher were

averaged and shown in the figure.





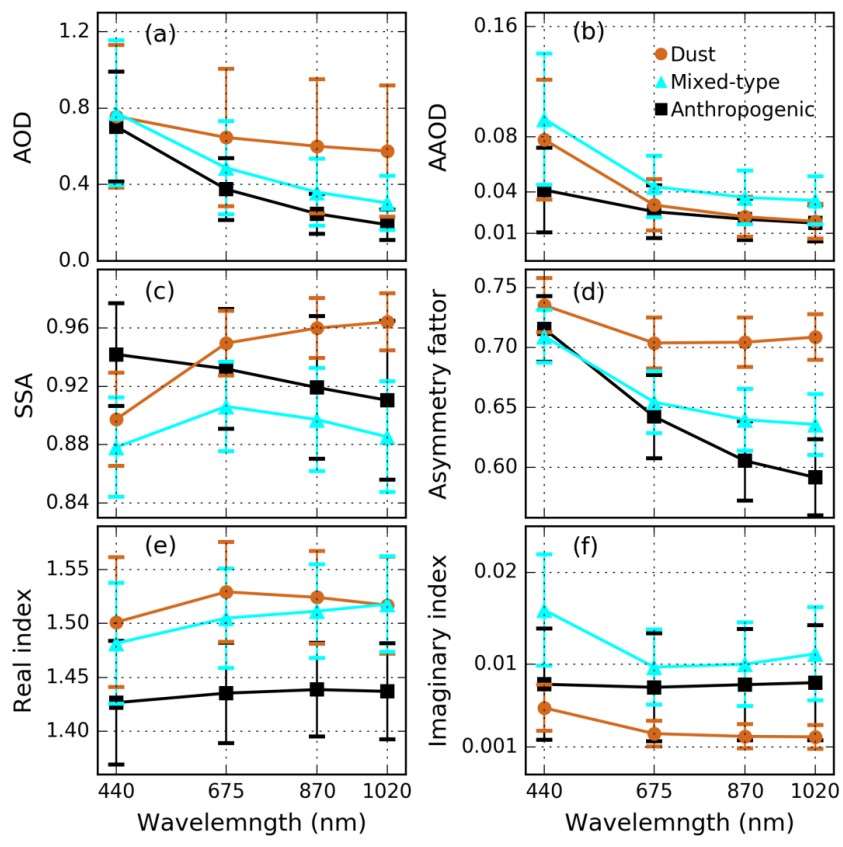


**Figure 3.** Spectral behavior of (a) AOD, (b) AAOD, (c) SSA, (d) asymmetry factor, (e) real part of

the complex refractive index, and (f) imaginary part of the complex refractive index for dust,

mixed-type and anthropogenic aerosols averaged from East Asian sites.





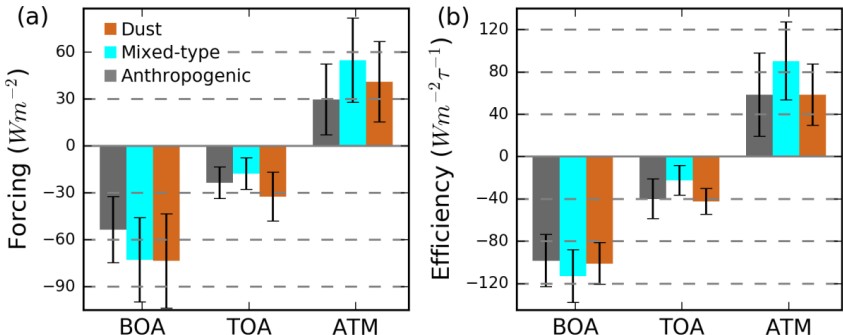


**Figure 4.** (a) Radiative forcing and (b) radiative efficiency of the dust, mixed-type, and

anthropogenic aerosols averaged from East Asian sites.







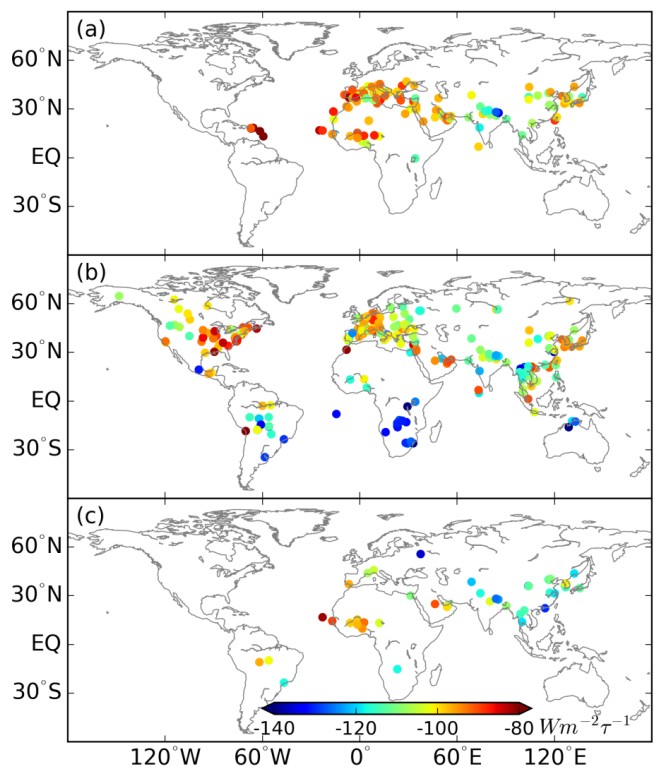


**Figure 5.** Aerosol radiative efficiency at BOA: (a) dust aerosols, (b) anthropogenic aerosols, and

(c) mixed-type aerosols.








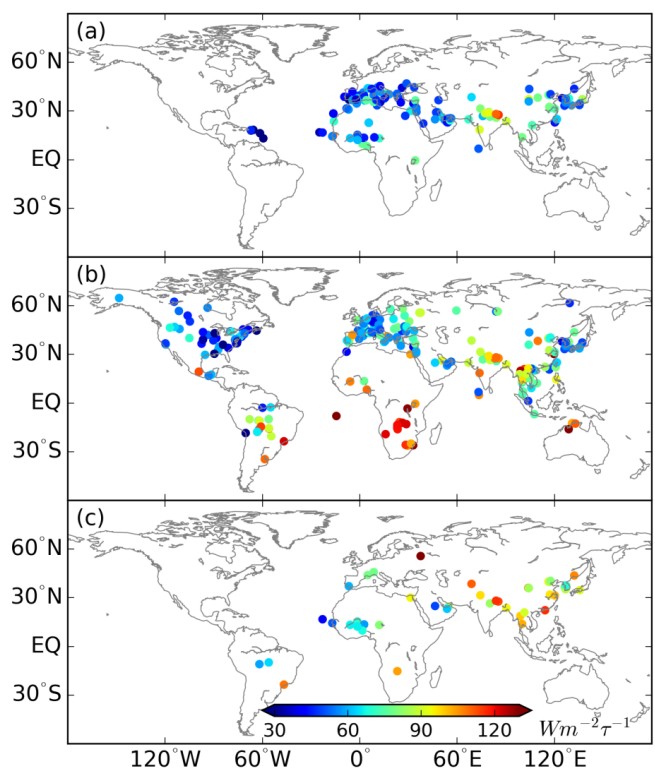


**Figure 6.** Aerosol radiative efficiency in ATM: (a) dust aerosols, (b) anthropogenic aerosols, and

(c) mixed-type aerosols.







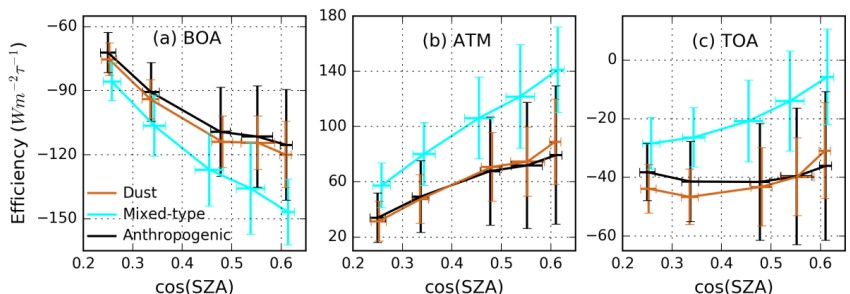


**Figure 7.** Aerosol direct radiative efficiency as a function of the cosine of solar zenith angle for

the dust, mixed-type and anthropogenic aerosols: (a) BOA, (b) ATM, and (c) TOA.





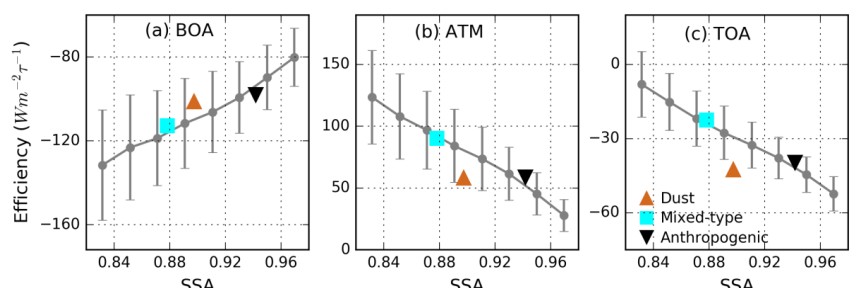

**Figure 8.** Aerosol direct radiative efficiency as a function of SSA for the dust, mixed-type and
anthropogenic aerosols: (a) BOA, (b) ATM, and (c) TOA.






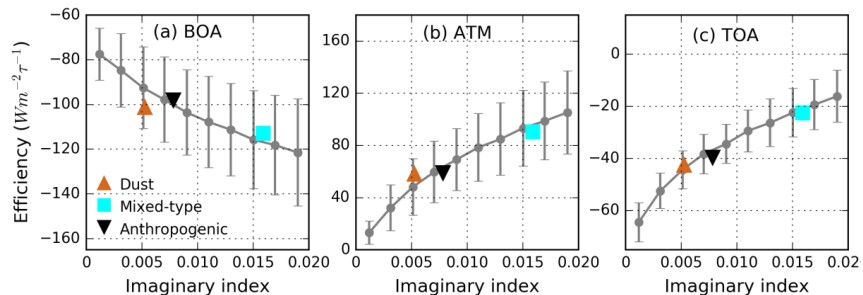


**Figure 9.** Aerosol direct radiative efficiency as a function of imaginary refractive index for the
dust, mixed-type and anthropogenic aerosols: (a) BOA, (b) ATM, and (c) TOA.






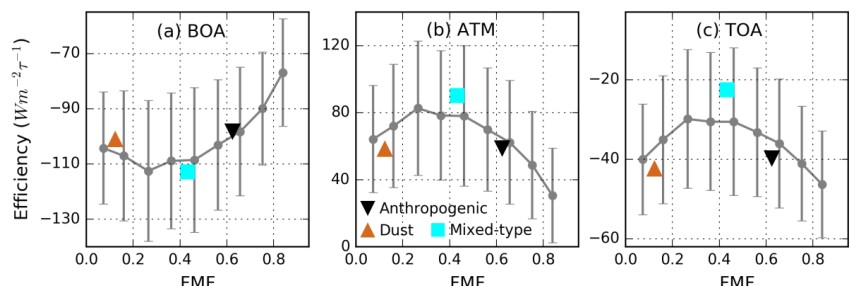


**Figure 10.** Aerosol direct radiative efficiency as a function of FMF in the East Asian region: (a)
BOA, (b) ATM, and (c) TOA. The average radiative efficiencies of the dust, mixed-type and
anthropogenic aerosols are also plotted.