# Peer review of "Radiative absorption enhancement of dust mixed with anthropogenic"

_Atmospheric Chemistry and Physics, 2018_

## Short Comment (SC1) · 31 Jan 2018

The authors should pay more attentions on the AERONET data usage: Please see: https://aeronet.gsfc.nasa.gov/new_web/data_usage.html

It highlights: "Please consult with the PI(s) of the data to be used. " "Please consider authorship for the PI(s) and/or the following acknowledgement:"

---

## Author Comment (AC1) · 15 Mar 2018

We would like to thank Dr. Wang for kindly reminding us of the AERONET data usage policy. We did acknowledge the AERONET data usage in the originally submitted manuscript.

———————————————

---

## Referee Comment (RC1) · Anonymous Referee #3 · 11 Apr 2018

Aerosol particles have been found to dramatically affect the weather and climate in East Asia, a hot spot region in term of dust and anthropogenic aerosol emissions. However, the effect of aerosol radiative enhancement due to the mixing of dust and anthropogenic aerosol remains to be poorly understood. Based on long-term AERONET observations along with radiative transfer model calculation, the mixed-type aerosols are found to exhibit a significantly larger BOA cooling radiative efficiency and ATM warming radiative efficiency compared with either dust or anthropogenic aerosols. This strong gradient of radiative effect in the vertical could be one of the factors explaining the deterioration of air quality in East Asia (including India and China). The paper is well written and structured. The classification method is robust by combining the SSA and angstrom coefficient. And the estimation of BOA radiative efficiency is much better

compared with previous methods through explicitly accounting for the nonlinear dependence between aerosol direct radiative forcing and AOD. Therefore, I recommend this paper be accepted for publication in ACP pending minor revision.

Specific Comments:

1. Line 93: "mixing" -> "mixed" 2. Lines 102-103: can "under different air quality conditions" be changed to "under pristine and polluted conditions"? 3. Lines 108-109: "the radiative absorption enhancement by the aerosol mixtures in East Asia has not been assessed." is not accurate since there is a lot of work involved in the radiative absorption enhancement, e.g., Cui et al. 2016 (doi: 10.1016/j.scitotenv.2016.02.026). 4. Line 240: Fig. 4a -> Fig.3a

––––––––––––––––––––––––––––

---

## Author Response (AR1)

**Author's Response of " Radiative absorption enhancement of dust mixed with anthropogenic pollution over East Asia" by Pengfei Tian et al.**

**Response to Referee #3**

Aerosol particles have been found to dramatically affect the weather and climate in East Asia, a hot spot region in term of dust and anthropogenic aerosol emissions. However, the effect of aerosol radiative enhancement due to the mixing of dust and anthropogenic aerosol remains to be poorly understood. Based on long-term AERONET observations along with radiative transfer model calculation, the mixed-type aerosols are found to exhibit a significantly larger BOA cooling radiative efficiency and ATM warming radiative efficiency compared with either dust or anthropogenic aerosols. This strong gradient of radiative effect in the vertical could be one of the factors explaining the deterioration of air quality in East Asia (including India and China). The paper is well written and structured. The classification method is robust by combining the SSA and angstrom coefficient. And the estimation of BOA radiative efficiency is much better compared with previous methods through explicitly accounting for the nonlinear dependence between aerosol direct radiative forcing and AOD. Therefore, I recommend this paper be accepted for publication in ACP pending minor revision.

**Response**: We are grateful to Referee #3 for the constructive and helpful comments. All the comments and concerns raised by the referee have been explicitly considered and incorporated into the revised manuscript. For clarity purpose, we have listed the reviewers' comments followed by our responses.

**Specific Comments:**

1. Line 93: "mixing" -> "mixed"

**Response**: Agreed and corrected in the revised manuscript (Line 94 in the revised manuscript; Line 143 in this author's response).

2. Lines 102-103: can "under different air quality conditions" be changed to "under pristine and polluted conditions"?

**Response**: Agreed and corrected in the revised manuscript (Line 104; Line 153).

3. Lines 108-109:"the radiative absorption enhancement by the aerosol mixtures in East Asia has not been assessed." is not accurate since there is a lot of work involved in the radiative absorption enhancement, e.g., Cui et al. 2016 (doi: 10.1016/j.scitotenv.2016.02.026).

**Response**: According to the reviewer's comment and our research, we replaced "the radiative absorption enhancement by the aerosol mixtures in East Asia has not been assessed." with a more proper description "further studies are urgently demanded to better understand the key role that the East Asian aerosol mixtures play in the formation mechanism of regional air pollution." (Lines 109-111; Lines 158-160). We cited Cui et al. 2016 (doi: 10.1016/j.scitotenv.2016.02.026) in the revised manuscript (Lines 65-66 and Lines 426-428; Lines 65-66 and Lines 475-477).

4. Line 240: Fig. 4a -> Fig.3a

**Response**: Agreed and corrected in the revised manuscript (Line 242; Line 291).

**Additional changes**

The funding of Pengfei Tian was included in the revised manuscript (Lines 394-395 in the revised manuscript; Lines 443-444 in this author's response), which was missing in the original submitted manuscript.

[revised manuscript text omitted]